# Toxic AGEs (TAGE) Cause Lifestyle-Related Diseases

**DOI:** 10.3390/antiox13111372

**Published:** 2024-11-09

**Authors:** Masayoshi Takeuchi

**Affiliations:** Department of Advanced Medicine, Medical Research Institute, Kanazawa Medical University, 1-1 Daigaku, Uchinada, Kahoku 920-0293, Ishikawa, Japan; takeuchi@kanazawa-med.ac.jp; Tel.: +81-76-218-8456

**Keywords:** advanced glycation end-products (AGEs), toxic AGEs (TAGE), lifestyle-related diseases (LSRD), TAGE degradation pathway, TAGE structures

## Abstract

Advanced glycation end-products (AGEs) play a role in the onset/progression of lifestyle-related diseases (LSRD), suggesting that the suppression of AGE-induced effects can be exploited to prevent and treat LSRD. However, AGEs have a variety of structures with different biological effects. Glyceraldehyde (GA) is an intermediate of glucose, and fructose metabolism and GA-derived AGEs (GA-AGEs) have been associated with LSRD, leading to the concept of toxic AGEs (TAGE). Elevated blood TAGE levels have been implicated in the onset/progression of LSRD; therefore, the measurement of TAGE levels may enable disease prediction at an early stage. Moreover, recent studies have revealed the structures and degradation pathways of TAGE. Herein, we provide an overview of the research on TAGE. The TAGE theory provides novel insights into LSRD and is expected to elucidate new targets for many diseases.

## 1. Introduction

Previous studies have revealed that advanced glycation end-products (AGEs) are one of the factors involved in the onset/progression of lifestyle-related diseases (LSRD). Therefore, suppressing the action of AGEs produced in the body may be effective in preventing and treating diseases; however, various AGEs are produced in the body [1,2]. Our group has revealed that toxic AGEs (TAGE), which are produced from glyceraldehyde (GA), an intermediate in sugar metabolism, are strongly related to LSRD, and we have proposed the concept of “TAGE theory” [1,3,4]. For example, the accumulation of TAGE is involved not only in diabetes mellitus (DM) and its complications, but also in various diseases such as non-alcoholic steatohepatitis (NASH), cardiovascular disease (CVD), Alzheimer’s disease (AD), cancer, and infertility [1,2,3,4].

The habitual intake of sugars, such as sucrose and high-fructose corn syrup (HFCS), and dietary AGEs (dAGEs) from processed foods/sugar-sweetened beverages (SSB), activates the sugar metabolism pathways (fructolysis, glycolysis, and polyol pathways) and causes excessive production of GA, an intermediate of glucose (Glu)/fructose (Fru) metabolism, in cells [2,3]. GA binds to intracellular proteins to generate and accumulate TAGE. This process causes cell damage and cell death due to a decrease in intracellular protein homeostasis (proteostasis) [3,4,5,6]. As a result, TAGE leak into the surrounding extracellular fluid and blood, and together with the action mediated by the TAGE-RAGE (receptor for AGEs) axis [1,2,3], they contribute to the onset/progression of LSRD [1,2,3,4]. Fluctuations in blood TAGE levels induced by various cell disorders are strongly correlated with the onset/progression of LSRD, including pre-disease states, potentially making TAGE a useful indicator for predicting diseases in the early stages [2,4]. Furthermore, TAGE levels are expected to be useful for selecting patients for predictive intervention in “preemptive medicine.” Recent studies have also revealed the structure of TAGE, showing that the structure differs from other GA-AGEs [7,8]. By leveraging this structural specificity, it may be possible to develop a therapeutic strategy targeting TAGE. Overall, minimizing the accumulation of TAGE produced in the body represents a novel strategy for the promotion of anti-aging, prevention of LSRD, and the extension of a healthy lifespan.

This review provides an overview of the research on the relationship between TAGE and LSRD.

## 2. Glycation and AGEs, the History of Glycation Research

To live a healthy life, we must consume the necessary nutrients in a balanced manner. In particular, a normal function cannot be maintained without a certain amount of Glu in the blood, but as a drawback, protein glycation (also known as the Maillard reaction) is constantly progressing in the body. Hemoglobin (Hb) A1c, which is commonly used in DM diagnosis, is an early glycation product generated when Glu binds to Hb in erythrocytes, and glycoalbumin is an early glycation product generated when Glu binds to albumin in the blood. If a state of hyperglycemia continues, the glycation reaction progresses further, and AGEs are generated [9,10,11,12] (Figure 1). AGEs endogenously/exogenously accumulate in the body [13,14,15]. Different types of AGEs are produced depending on the sugars and dicarbonyl compounds involved [16,17,18,19,20,21,22].

The history of research into protein glycation began in 1912 with the discovery of melanoidin by Maillard [23]. At the time, this reaction was considered important in the field of food chemistry, known as the “browning reaction,” and it was studied as a phenomenon related to the taste and texture of food. More than half a century after the discovery of melanoidin, in 1968, early glycation products like HbA1c were found in the body [24], and in 1986, late glycation products (AGEs) were discovered [25]. In 1992, RAGE was cloned as one of the AGE receptors [26], and in 2001, Yamamoto et al. created a genetically modified animal [27]. A variety of AGE structures have been reported, from simple structures, such as N^ε^-(carboxymethyl)lysine (CML), to more complex structures, namely pyrraline and pentosidine [28,29,30], and more than 30 types of structures have been reported to date [31,32,33,34]. Moreover, in 1996, a paper was published stating that most of the known anti-AGE antibodies recognized the CML structure [35], and the concept that “CML is the main structure of AGEs” has spread worldwide. However, by producing and studying various anti-AGE antibodies, our group realized that other AGE structures (non-CML AGEs) were more correlated with clinical parameters [1,2]. Additionally, we found non-CML AGEs that showed strong cytotoxicity (TAGE) and published the “TAGE (toxic AGEs) hypothesis in various chronic diseases” in 2004 [1].

## 3. Overview of AGEs Formation Pathway in the Body

In the body, AGEs were thought to be mainly generated by the reaction of Glu with proteins, known as Glu-derived AGEs (Glu-AGEs). Most cyclic Glu and Fru structures do not undergo glycation reactions, but when they form chain structures, the aldehyde groups bind to amino groups in proteins, and glycation reactions proceed. However, because the proportion of chain structures is extremely low, it takes several days to generate HbA1c, an early glycation product, and several weeks to several months to generate Glu-AGEs, a late glycation product derived from Glu (Figure 1). The rate of glycation differs according to the types of sugars involved, with GA being faster in glycation reactions than Glu and Fru [36,37,38,39]. In other words, humans have wisely chosen Glu as an energy source because it is the least likely to naturally undergo glycation reactions.

In recent years, it has been revealed that AGEs are also generated from metabolic intermediates of Glu and Fru, as well as Maillard reaction intermediates. Seven different classes of AGEs (glyoxal (GO)-, glycolaldehyde (Glycol)-, methylglyoxal (MGO)-, 3-deoxyglucosone (3-DG)-, Glu-, Fru-, and GA-AGEs), as well as CML forms, have been detected in the serum of hemodialysis (HD) patients with diabetic nephropathy (DN) (DN-HD) [1,2] (Figure 2).

When GA, an intermediate in sugar metabolism, is produced, it instantly binds to proteins to generate GA-AGEs. Thus, it is feared that modern eating habits that involve a high intake of HFCS may promote the production of GA-AGEs. The following three pathways are known to produce GA, the precursor of GA-AGEs, in vivo [3,8] (Figure 2).

(i)GA is produced by nonenzymatic dephosphorylation of glyceraldehyde-3-phosphate (GA-3-P), a glycolytic intermediate. Glycolysis is a fundamental metabolic pathway involving the conversion of Glu to pyruvate. When GA-3-P metabolism shifts to another route, the amount of GA increases, promoting the formation of GA-AGEs.(ii)GA is directly produced from Fru by the action of fructokinase (FK) and aldolase B, which are key enzymes in fructolysis [40,41]. FK accumulates in the liver after a meal, and Fru is phosphorylated by a specific FK to form Fru-1-phosphate (F-1-P), which is then cleaved by aldolase B, producing dihydroxyacetone-phosphate (DHA-P) and GA.(iii)GA is produced by fructolysis but from Fru produced in the polyol pathway [42,43], an alternative pathway of glycolysis, under hyperglycemic conditions. Aldose reductase catalyzes the reduction of Glu to sorbitol, which is then converted to Fru by sorbitol dehydrogenase. Under hyperglycemic conditions, Glu concentrations are elevated in insulin-independent tissues, enhancing the activity of the polyol pathway.

Sugars have been correlated with numerous risk factors, including those for obesity and chronic diseases such as NASH, DM, and CVD. Major sources for GA-AGEs are sugar- and starch-rich foods like rice, bread, and SSB (i.e., with added sucrose or HFCS), which have been associated with higher colorectal cancer (CRC) mortality [44]. GA-AGEs are generated by the habitual excessive intake of sugars and dAGEs and accumulate in various cells that compose the liver, heart, and brain [2].

MGO is formed also as a byproduct of glycolysis, in which the intermediate GA-3-P and DHA-P, spontaneously degrade to generate MGO [45,46] (Figure 2). MGO is an Arg-directed glycating agent that mainly forms MGO-derived hydroimidazolone 1 (MG-H1) [47,48] and argpyrimidine (ArgP) [49,50].

## 4. Origin of the Name TAGE

When various AGEs were artificially produced in a test tube and added to primary cultured neurons derived from the cerebral cortex of fetal rats, neuronal cell death was observed. Notably, neuronal cell death induced by the addition of GA-AGEs, which had the strongest cytotoxicity, was only suppressed when anti-GA-AGE antibody was added [3].

Furthermore, when an AGE fraction obtained from the blood of DN-HD patients, which contained large amounts of AGEs, was added to primary cultured neurons, neuronal cell death was reproduced, and when various anti-AGE antibodies were added at the same time, cell death was suppressed only when the anti-GA-AGE antibody was added [3]. We conducted in vitro experiments to examine the binding of the seven distinct classes of AGEs and CML, which were identified in blood collected from DN-HD patients [1,2], to RAGE using a purified human RAGE protein and found that the dissociation constant for GA-AGEs was 0.36 μM. Similar binding kinetics were noted in cellular assays using COS-7 cells expressing RAGE [1,3]. Additionally, accumulation of GA-AGEs was observed in neurons in the lesioned areas of the brains of AD patients [3], and it was shown that GA-AGEs, among the various AGEs present in the body, had adverse effects on the body; hence, they were named TAGE to distinguish them from other AGEs [1,2,3,4].

These results suggest that AGE structures containing epitopes recognized by anti-GA-AGE antibody is relatively toxic. Therefore, specific AGEs that bind to anti-GA-AGE antibody is TAGE, which can be distinguished from GA-AGEs such as trihydroxy-triosidine [51], GA-derived pyridinium compound (GLAP) [52], GA-derived pyrrolopyridinium lysine dimers (PPGs) [53], MG-H1 [54], and ArgP [55], as well as other AGEs [1,2] (Figure 2).

## 5. The TAGE-RAGE Axis

Receptor-dependent or -independent mechanisms influence cellular dysfunction and tissue damage caused by AGEs. Different classes of biological reactions mediated by RAGE have been examined, and several AGE-binding proteins were identified [33]. RAGE regulatory fragments, such as a soluble form of RAGE and endogenous secretory RAGE, which functions as a decoy receptor of RAGE, play important roles in pathobiology [56,57]. RAGE is normally expressed in a variety of cells, including hepatocytes, cardiomyocytes, endothelial cells, pericytes, and neurons [1,3,6]. Most healthy adult tissues express low levels of RAGE; however, its expression is up-regulated under pathological conditions, as observed in DM, CVD, and cancer [1,6]. Several AGE receptors, namely AGE-R1/-R2/-R3 and the scavenger receptor family (classes A, B, E, and H), among others, and stabilin-1/-2, which have the opposite function to RAGE, are essential for maintaining AGE homeostasis [33].

AGEs have been implicated in the etiology of multiple chronic diseases, including cancers, due to their pro-inflammatory and pro-oxidative properties. The binding of AGEs to RAGE was shown to increase oxidative stress and inflammation by generating reactive oxygen species (ROS), which activates multiple signaling pathways involved in cell proliferation and apoptosis, including the PI3K/Akt, NF-κB, and transforming growth factor (TGF)-β pathways [58,59]. The AGEs-RAGE activation of these downstream signaling pathways may be the main mechanism for the promotion of chronic diseases by AGEs [56,57].

Among various AGE subgroups, TAGE have the strongest binding affinity for RAGE, leading to potentially stronger downstream effects and enhanced activation of signaling pathways. In addition, the binding of TAGE to RAGE was shown to induce cell damage, pro-inflammatory cytokine production, and vascular endothelial growth factor (VEGF) expression in various cells, all of which may contribute to the onset/progression of LSRD [1,3].

## 6. Structures and Formation Pathway of TAGE

GA-AGEs are a structurally heterogeneous group of molecules, but the specific structure that causes their toxicity has recently become clear. The GA-AGE structures include triosidines, GLAP, PPGs, MG-H1, and ArgP (Figure 3). Trihydroxy-triosidine, GLAP, and PPGs are generated by the reaction of GA with Lys residues in proteins. MG-H1 and ArgP are generated by the reaction of GA with Arg residues, but it has been reported that ArgP is only formed when Lys residues coexist with Arg residues. Lys residues were significantly more likely to form GA-AGEs than Arg residues, which may be important for the development of novel strategies to prevent and treat GA-AGE-dependent diseases [60]. Notably, anti-TAGE antibody does not recognize known GA-AGE structures that contain pyridinium or pyrrolopyridinium rings, such as trihydroxy-triosidines, GLAP, and PPGs.

### 6.1. Formation Pathways of Known GA-AGEs and TAGE

Figure 3 shows the formation pathways of the trihydroxy-triosidine, GLAP, and PPG structures. In the Maillard reaction, a Schiff base is first formed between the carbonyl group of GA and the amino group (P-NH_2_) in the protein, followed by the Amadori rearrangement to generate a ketoamine (adduct 1). Then, the imino group of adduct 1 reacts with the carbonyl group of another molecule of GA to form the PPG-1 and PPG-2 structures with pyrrolopyridinium rings [53], or the triosidine and GLAP structures with pyridinium rings [51,52,61], which have been reported as known GA-AGE structures.

The formation pathway of TAGE is also shown in Figure 3. In the first reaction, a Schiff base is formed between GA and P-NH_2_, similar to the known GA-AGE structures (trihydroxy-triosidine, GLAP, and PPGs), followed by an Amadori rearrangement to generate adduct 1. Next, a second molecule of P-NH_2_ reacts with the carbonyl group of adduct 1 to form adduct 3. It was then predicted that GA would further react to generate 1,4-dihydropyrazine compounds by following route A (dimeric structure) or route B (trimeric structure) [7].

### 6.2. Structure of TAGE

Yamamoto et al. isolated and identified two compounds with a 1,4-dihydropyrazine ring from the reaction product of N^α^-carbobenzoxy-L-lysine (CBZ-Lys) and GA [WO2020045646A1]. We considered the two compounds with GA-derived 1,4-dihydropyrazine rings that form intramolecular and intermolecular crosslinks to be TAGE structures (Figure 4A). After reacting proteins (lysozyme and ribonuclease A) with GA for a certain period, the proteins were stained with Coomassie blue, and monomer, dimer, and trimer bands were detected (Figure 4B, left). In addition, a similar band pattern was detected by Western blotting using an anti-TAGE antibody (Figure 4B, right). These results suggest that the GA-AGEs derived from lysozyme and ribonuclease A, generated in vitro, have a structure with a 1,4-dihydroprazine backbone that forms intramolecular and intermolecular crosslinks.

## 7. TAGE as a Cytotoxic Factor

By examining the relationship between intracellular TAGE accumulation and cytotoxicity, it has become clear that the generation and accumulation of TAGE causes cytotoxicity not only in neurons but also in liver cells and cardiac muscle cells.

### 7.1. Association with Neuronal Damage

Given that TAGE are generated inside cells, the relationship between the addition of GA, a TAGE precursor, and neuronal damage was examined. As a result, (i) neuronal cell death was caused by intracellular TAGE accumulation, and pathological changes characteristic of AD, such as a decrease in amyloid β protein and an increase in phosphorylated tau protein, were observed [3]. Furthermore, (ii) the cytoskeletal protein β-tubulin was significantly modified by TAGE, causing abnormal polymerization of microtubule formation and inhibiting axonal elongation [3,5], and finally, (iii) TAGE accumulation in glial cells may be involved in the breakdown of the blood-brain barrier by inducing cell death.

### 7.2. Association with Hepatocellular Damage

Regarding the relationship with hepatocellular damage, (i) TAGE accumulation in hepatic parenchymal cells causes mitochondrial dysfunction, enhances the production of ROS, and induces inflammation [3]. Moreover, (ii) intracellular TAGE accumulation induces modification of molecular chaperones, such as heat shock cognate (HSC) 70 and caspase-3, inducing necrotic cell death [3,6]. Finally, (iii) TAGE leak out of the cells and affect surrounding hepatic parenchymal cells, stellate cells, and cancer cells through RAGE, progressing to hepatitis, liver fibrosis, and liver cancer [3]. ROS have also been shown to up-regulate RAGE expression and increase TAGE production, which may play a role in the onset/progression of NASH.

### 7.3. Association with Cardiomyocyte Damage

Regarding the association with cardiomyocyte damage, (i) TAGE accumulation in cardiomyocytes reduces the cardiac rate and induces cell death by inhibiting the autophagy function [3], (ii) TAGE modification of heat shock protein (HSP) 90β may be involved in this autophagy activity, and (iii) TAGE accumulation in surrounding cardiac fibroblasts, which are involved in protecting the myocardium, may cause fibroblast death and reduce myocardial protection [4].

### 7.4. Summary

In addition, it has been revealed that intracellular TAGE accumulation also causes cell damage in pancreatic β cells, myoblasts, osteoblasts, and pancreatic ductal epithelial cells [3,4].

Therefore, we investigated the possibility of using TAGE, which leak into the blood in association with various cell disorders, as a diagnostic marker.

## 8. Blood TAGE as a Predictive Marker for LSRD

The amount of TAGE in blood was measured by a competitive enzyme-linked immunosorbent assay (ELISA) method using a TAGE-specific antibody developed by us. It has been shown that this antibody is a TAGE structure-specific antibody that does not recognize known AGE structures such as CML, triosidines, GLAP, MG-H1, ArgP, etc., [1,2].

### 8.1. Association with NASH

We investigated the relationship between NASH, a liver manifestation of metabolic syndrome (MetS), and the amount of TAGE in the blood. As a result, we found that (i) TAGE levels are significantly higher in NASH patients than in healthy individuals and non-alcoholic fatty liver (NAFL) patients. TAGE accumulation is observed in liver tissue, and if the cut-off value is set at 8.53 U/mL, it is possible to distinguish NASH from NAFL with a certain degree of probability; (ii) TAGE levels are positively correlated with the homeostasis model assessment of insulin resistance (IR) (HOMA-IR), an index of IR, and negatively correlated with adiponectin levels; (iii) liver function is restored with a decrease in TAGE levels when NASH patients with lipid metabolism disorders are treated with atorvastatin; and (iv) the TAGE levels were even higher in patients with non-B or non-C type hepatocellular carcinoma (NBNC-HCC) than in patients with NASH [2,4]. Overall, the fluctuation in blood TAGE levels may be a useful biomarker for the prevention, diagnosis, evaluation, and treatment of NASH.

### 8.2. Association with CRC

The European Prospective Investigation into Cancer and Nutrition cohort study, conducted mainly in Europe and the United States, revealed that the risk of developing rectal cancer after four years was approximately doubled in those with high TAGE levels [2,4]. It was also shown that those who drink alcohol have a higher risk of developing rectal cancer. This increased risk due to alcohol consumption was observed for an intake similar to that observed in NASH (less than 30 g/day for men and 20 g/day for women), and thus NASH patients with high blood TAGE levels may have a higher risk of rectal cancer, in addition to liver cancer.

Furthermore, an observational study was conducted on the relationship between blood TAGE levels and mortality, and the blood TAGE levels before CRC diagnosis were correlated with both CRC-specific mortality and all-cause mortality. In particular, a stronger correlation was shown between TAGE levels and CRC-specific mortality in rectal cancer patients [62].

### 8.3. Association with CVD and Heart Failure

Elevated blood TAGE levels are seen even in healthy individuals whose blood test values are all within the normal range, and since a decrease in the number and function of endothelial progenitor cells is observed in groups with high blood TAGE levels, TAGE may be a marker for predicting the future progression of arteriosclerosis and CVD [2]. Additionally, in circulatory outpatients, the localization of plaque by carotid artery ultrasound coincided with the uptake of [^18^F] fluorodeoxyglucose, an indicator of inflammation evaluated by positron emission tomography. Furthermore, a correlation was observed between the blood TAGE levels and the degree of vascular inflammation indicated by the target-to-background ratio, showing that TAGE may be a biomarker of vascular inflammation in arteriosclerotic lesions [2].

Recently, the TAGE and tumor necrosis factor (TNF)-α levels showed close associations with left ventricular ejection fraction and brain natriuretic peptide values in patients with diabetic adverse cardiac remodeling [4]. TAGE and TNF-α may play a pathological role in the development of diabetic adverse cardiac remodeling.

### 8.4. Association with Infertility Treatment

Investigations into the relationship between blood TAGE levels and the number of eggs collected and continued pregnancy rates in infertility treatment patients who were considered to be in a pre-disease state revealed that both factors decreased in proportion to age, but that the continued pregnancy rate was poor in groups with high blood TAGE levels, even in younger women [2]. In response, poor ovarian responders who had been unable to become pregnant were given sitagliptin, a dipeptidyl peptidase (DPP)-4 inhibitor, along with advice on improving their dietary habits and were then re-administered the infertility treatment. As a result, the continued pregnancy rate significantly improved as the TAGE levels decreased.

### 8.5. Summary

As described above, it is clear that changes in blood TAGE levels can be used as a biomarker for predicting the onset/progression of LSRD, including pre-disease, regardless of whether the patient has DM or not [2,4], which may also help in the selection of patients for predictive intervention in “preemptive medicine”.

## 9. Strategies to Prevent LSRD by Suppressing TAGE

Habitual excessive intake of foods and beverages high in sugar and HFCS not only causes obesity and MetS but also increases the risk of LSRD such as NASH, DM, and CVD. The American Heart Association and the World Health Organization (WHO) have announced guidelines stating that “to maintain a healthy lifestyle, the daily sugar (i.e., sucrose and HFCS) intake should be limited to less than approximately 25 g for adults [63,64,65].” Moreover, according to a review paper published by Huang et al. in 2023, to reduce the harmful effects of sugar on health, it is recommended that “in addition to the WHO guidelines, SSB intake should be limited to less than 200–355 mL/week” [66].

### 9.1. Sugar Intake and TAGE

Sugars have been associated with a variety of risk factors, mainly including obesity and chronic diseases such as NASH, CVD, DM, and some cancers [44,67,68,69,70,71,72,73,74]. When the sugar content of beverages was measured, many beverages on the market exceeded the recommended daily limit, and the beverages with the highest sugar content contained nearly 60 g per 500 mL. When normal rats were given 10% HFCS 55 (Fru 55%/Glu 45%), using the same concentration as these beverages, the amount of TAGE in the blood increased in proportion to the amount of TAGE accumulated in the liver [75].

Conversely, when acarbose, an α-glucosidase inhibitor, was administered to type 2 DM patients, there was no change in HbA1c values, and the amount of TAGE decreased with the improvement of postprandial hyperglycemia [2].

Ultimately, it is clear that habitual and excessive intake of Fru and Glu causes the accumulation of TAGE in the body.

### 9.2. dAGEs Intake and TAGE

A variety of precursors and complex mechanisms of AGE formation generate many chemically diverse AGE molecules that have been identified in foods [76,77,78,79,80,81,82]. Most often, processed foods in modern diets are rich in sugars and proteins that undergo the Maillard reaction during thermal processing, leading to AGE formation [83,84,85,86]. Glycation reactions occur during cooking and manufacturing processes in foods/beverages that contain sugars, producing large amounts of Glu-AGEs and Fru-AGEs, but no TAGE [2]. When a high-AGE beverage was administered to normal rats, it was shown to not only cause the accumulation of Glu-AGEs in the liver but also increase the expression of RAGE and VEGF, along with the accumulation of TAGE, which is not contained in the beverage [2].

In contrast, the administration of the oral charcoal adsorbent Kremezin to non-DM patients with chronic kidney disease reduced the amounts of Glu-AGEs contained in foods and beverages, as well as the amount of TAGE in the blood [2]. In addition, Japanese cuisine not only has low amounts of AGEs but also uses many ingredients that contain dietary fiber, which inhibits the digestion and absorption of sugar. Moreover, ingredients that are high in insoluble dietary fiber have been shown to have a stronger dAGE adsorption effect than Kremezin [unpublished data].

In other words, habitual and excessive intake of dAGEs promotes the accumulation of TAGE in the body and is involved in the onset/progression of LSRD.

### 9.3. Summary

Research has revealed that TAGE accumulate in the body due to the habitual and excessive intake of sugar, HFCS, and dAGEs, which are characteristic of modern diets. This accumulation is involved in the onset/progression of LSRD, making TAGE a potential target for preventing LSRD and extending a healthy lifespan (Figure 5).

The “new dietary strategy for realizing a healthy and long-lived society” is based on regular eating habits; it is recommended to (i) avoid the habitual intake of processed foods and beverages, which can cause TAGE accumulation if consumed in excess, (ii) support the regular intake of Japanese food, which is low in AGEs and uses many ingredients that suppress TAGE accumulation, and (iii) consider “the order in which you eat,” such as a “vegetable-first meal” [87,88,89] or a “carbohydrates-last meal” [90,91,92], which may suppress TAGE accumulation [93].

## 10. Degradation Pathway of TAGE

As mentioned above, GA-AGE structures containing epitopes recognized by anti-TAGE antibody may be cytotoxic, and intracellular TAGE accumulation leads to increased oxidative stress (ROS generation) and DNA damage response, disrupting normal protein homeostasis (proteostasis) [3,6]. Previous studies have shown that abnormal intracellular TAGE accumulation correlates with cell death in hepatocytes, neurons, cardiomyocytes, and other cells [3,4,5,6]. However, the molecular mechanisms for appropriately degrading and removing TAGE generated and accumulated in cells and maintaining proteostasis remain largely unknown.

Recently, our group reported on the existence of a degradation pathway for TAGE-modified proteins in an experimental model of TAGE formation using cultured cells [94]. Regarding the truncated forms (CHK1-CPs) and inactive mutant (d270KD) proteins of checkpoint kinase 1 (CHK1), an enzyme related to DNA damage response, the following findings were reported: (i) GA treatment induces TAGE modification and ubiquitin modification, which rapidly degrades CHK1-CPs and d270KD in cells; (ii) the inhibition of proteasome activity completely suppresses these degradation effects and causes accumulation of high-molecular-weight (HMW) d270KD protein; and (iii) the HMW complex contains p62/SQSTM1, which is known to be involved in selective autophagy. These results suggest that there is an intracellular system that detects slight structural changes in CHK1-CPs and d270KD caused by TAGE modification, which induces specific ubiquitin modification and degrades them through the proteasome pathway. Furthermore, the HMW TAGE-modified d270KD complex may be transported to the autophagy pathway by binding to p62, which targets the ubiquitin modification [94] (Figure 6).

In addition to the relationship between TAGE accumulation and disease, future research should focus on identifying ubiquitin modification sites caused by structural changes due to TAGE modification in individual intracellular proteins and discovering endogenous factors involved in their modification and degradation. Progress in this field is expected to support the development of effective intervention methods for diseases associated with the aggregation and accumulation of TAGE-modified proteins, including DM, NASH, CVD, and AD.

## 11. Therapeutic Drugs of TAGE in LSRD

High blood TAGE levels predict the onset/progression of LSRD, even in healthy subjects with normal blood test values.

(i)When healthy individuals took collagen tripeptide [95,96,97], which has TAGE inhibitory effects, the blood TAGE levels as well as the cardio–ankle vascular index decreased, which indicates the stiffness of the blood vessel walls. It is expected that an “improvement of TAGE levels will restore the elasticity of blood vessels and prevent arteriosclerosis” [4].(ii)When water chestnut (Trapa bispinosa Roxb.) extract [98,99,100] was administered to elderly patients with intractable infertility, the birth rate significantly increased with the decrease in blood TAGE levels [4]. Water chestnut extract significantly enhanced oocyte developmental potential, improved endometrial receptivity in natural cycles, and decreased blood and follicular fluid TAGE levels [4].(iii)Blood TAGE levels in NASH patients with dyslipidemia were reduced by treatment with atorvastatin. A 6-month treatment with atorvastatin decreased the activities of liver alanine aminotransferase and γ-glutamyl transpeptidase in all patients. Moreover, plasma adiponectin levels increased, and plasma TNF-α levels decreased in NASH and NAFL patients, while blood TAGE levels decreased [4].(iv)We demonstrated significant reductions in blood TAGE levels in DM patients treated with acarbose for 12 weeks [2]. We also found that blood TAGE levels were significantly reduced by a DPP-4 inhibitor, sulfonylurea, and insulin, and these decreases were associated with reductions in the biomarker levels of organ damage in DM patients [2,4]. Furthermore, we observed decreases in blood TAGE levels in DM patients treated with atorvastatin [2,4].(v)The administration of Kremedin, an oral adsorbent of dAGEs, decreased blood TAGE levels in non-diabetic patients with chronic renal failure [2,4].(vi)Yamamoto et al. found that an anti-TAGE monoclonal antibody inhibited eye angiogenesis in diabetic mice [WO2020045646A1].(vii)We recently reported on the protective role of pyridoxamine, which has TAGE inhibitory effects against the GA-mediated suppression of axonal outgrowth in zymosan-induced axonal elongation following nerve injury in mice [5].

These findings indicate the potential of using blood TAGE levels as a new biomarker for the early diagnosis of LSRD or the assessment of therapeutic strategies to prevent and treat the onset/progression of LSRD, regardless of the presence of DM.

## 12. Conclusions

TAGE are generated and accumulated in various cells that compose the liver, heart, brain, etc., as a result of the habitual intake of excessive amounts of sugars and dAGEs, causing various cell disorders that are involved in the onset/progression of LSRD. Thus, blood TAGE levels are strongly related to the onset/progression of LSRD. The results of many reports suggest that blood TAGE levels can be used to predict the onset/progression of diseases at an early stage in select patients for predictive intervention in “preemptive medicine”. Additionally, recent studies have revealed the structure of TAGE, which differs from other GA-AGEs. It may be possible to develop a treatment strategy that targets TAGE by taking advantage of this structural specificity.

In the future, it is necessary to clarify the relationships between the dynamics of TAGE and cell damage, cell aging, cell death, etc., focusing on the proteostasis of intracellular TAGE. It is also important to identify food ingredients that have TAGE structure-specific production-inhibiting/degradation-promoting effects and demonstrate their ameliorative effects. Accordingly, this is expected to promote the creation of innovative preventive measures for LSRD.

## Figures and Tables

**Figure 1 antioxidants-13-01372-f001:**
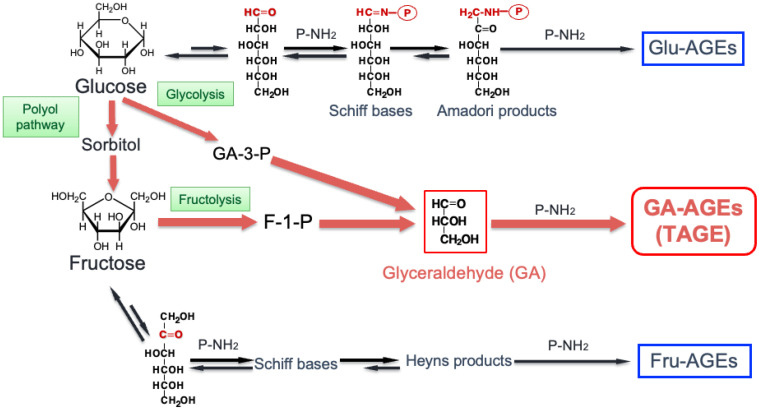
Glycation reaction (Maillard reaction).

**Figure 2 antioxidants-13-01372-f002:**
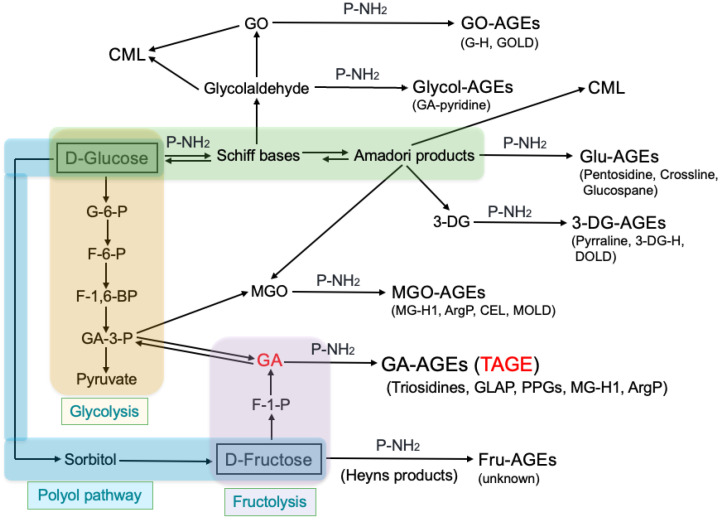
Overview of the various AGE production pathways in the body and major AGE structures. G-6-P, glucose-6-phosphate; F-6-P, fructose-6-phosphate; F-1,6-BP, fructose-1,6-bisphosphate; GA-3-P, glyceraldehyde-3-phosphate; F-1-P, fructose-1-phosphate; GO, glyoxal; GO-AGEs, glyoxal-derived AGEs; Glycol-AGEs, glycolaldehyde-derived AGEs; CML, N^ε^-(carboxymethyl)lysine; Glu-AGEs, glucose-derived AGEs; 3-DG, 3-deoxyglucosone; 3-DG-AGEs, 3-deoxyglucosone-derived AGEs; MGO, methylglyoxal; MGO-AGEs, methylglyoxal-derived AGEs; GA, glyceraldehyde; GA-AGEs, glyceraldehyde-derived AGEs; Fru-AGEs, fructose-derived AGEs; G-H, GO-derived hydroimidazolone; GOLD, GO-lysine dimer; 3-DG-H, 3-DG-derived hydroimidazolone; DOLD, 3-DG-lysine dimer; MG-H1, MGO-derived hydroimidazolone 1; ArgP, argpyrimidine; CEL, N^ε^-(carboxyethyl)lysine; MOLD, MGO-lysine dimer; TAGE, toxic AGEs; GLAP, glyceraldehyde-derived pyridinium; PPG, pyrrolopyridinium lysine dimer derived from glyceraldehyde.

**Figure 3 antioxidants-13-01372-f003:**
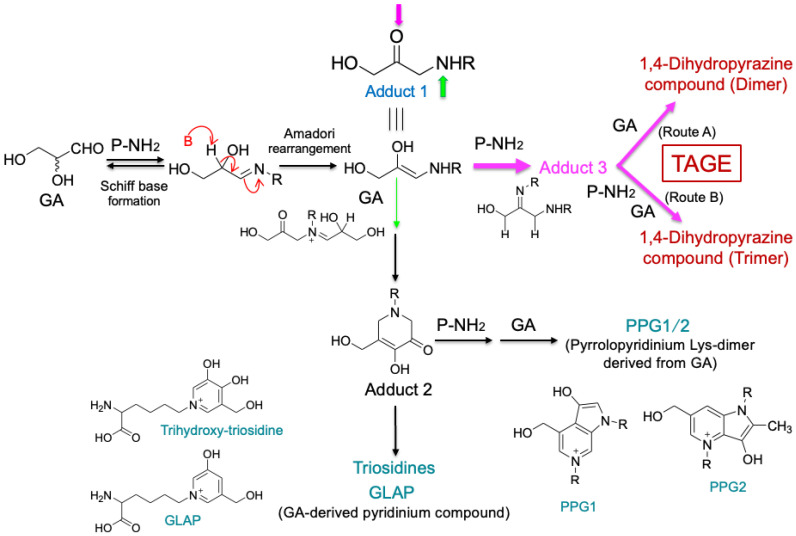
Overview of the known GA-AGE and TAGE formation pathways. GA, glyceraldehyde; P-NH_2_, a free amino residue of a protein; GLAP, glyceraldehyde-derived pyridinium; PPG, pyrrolopyridinium lysine dimer derived from glyceraldehyde; TAGE, toxic AGEs.

**Figure 4 antioxidants-13-01372-f004:**
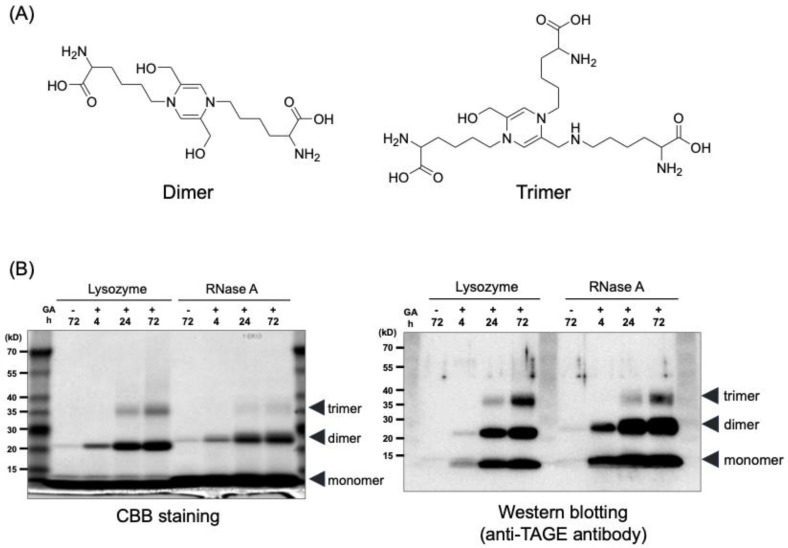
Estimated TAGE structures. (**A**) Dimer structure, 1,4-di(5-amino-5-carboxypentyl)-2,5-dihydroxymethyl-1,4-dihydropyrazine; Trimer structure, 1,4-di(5-amino-5-carboxypentyl)-5-(5-amino-5-carboxy-pentylaminomethyl)-2-hydroxymethyl-1,4-dihydropyrazine. (**B**) Left, Coomassie brilliant blue (CBB) staining; Right, Western blot analysis using an anti-TAGE antibody.

**Figure 5 antioxidants-13-01372-f005:**
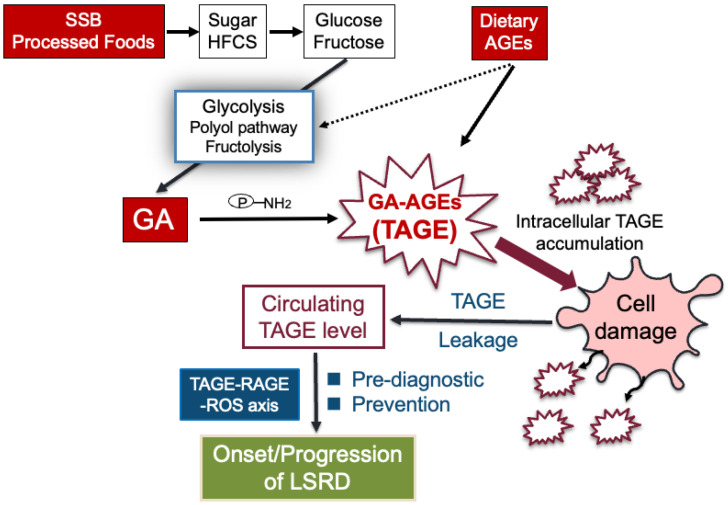
TAGE theory in LSRD. Habitual intake of sugar/HFCS and dietary AGEs causes excessive production of intracellular GA by activating the Glu/Fru metabolic pathway. GA binds to intracellular proteins to generate and accumulate TAGE, a type of GA-AGEs, which causes cell damage and cell death due to a decrease in proteostasis. As a result, TAGE leak out of cells and contribute to the onset/progression of LSRD, coupled with the action of the RAGE. Furthermore, since fluctuations in the blood TAGE levels are strongly correlated with the onset/progression of LSRD, including pre-disease states, it is expected that the onset of diseases can be predicted. This insight may support the prevention of LSRD and the extension of a healthy lifespan. SSB, sugar-sweetened beverages; HFCS, high-fructose corn syrup; AGEs, advanced glycation end-products; GA, glyceraldehyde; TAGE, toxic AGEs; RAGE, receptor for AGEs; ROS, reactive oxygen species; LSRD, lifestyle-related diseases; P-NH_2_, free amino residue of protein.

**Figure 6 antioxidants-13-01372-f006:**
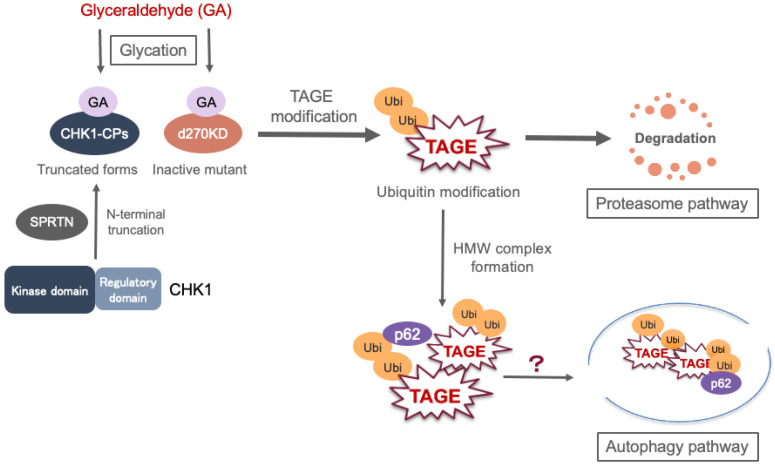
Overview of the TAGE degradation pathway. GA, glyceraldehyde; TAGE, toxic AGEs; Ubi, ubiquitin; CHK1, checkpoint kinase 1; SPRTN, Spartan protease; p62, p62/SQSTM1.

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
