# Peer review of "Toxic AGEs (TAGE) Cause Lifestyle-Related Diseases"

_antioxidants, 2024, doi:10.3390/antiox13111372_

Round 1
Reviewer 1 Report
Author has presented an overview of the research on TAGE mostly from the formation perspective and its impact on onset/progression of lifestyle-related diseases (LSRD). While the review is well written, there are some important aspects that need to be considered by the author for a more comprehensive review.
1. Authors needs to describe mechanisms by which TAGE drives LSRD. Does TAGE require engagement with its receptor to mediate cellular and molecular events to drive LSRD? Does receptor fo advanced glycation endproduct mediate some of TAGE actions in LSRD?
2. Figure 1 shows that polyol pathway (aldose reductase pathway) is, in part, responsible for generation of TAGEs. Authors should discuss preclinical and clinical studies that may have shown that blocking this pathway reduces TAGE and consequent LSRD.
3. Are TAGE degradation pathways altered in LSRD? Perhaps, a brief description on the balance between TAGE production vs degradation in the context of LSRD would be helpful.
Nothing specific to report.
Author Response
I would like to thank the Reviewers for their careful consideration of my manuscript. The comments were very helpful and I hope that my point-by-point responses are satisfactory, as shown in red. Moreover, the revised manuscript has been proofread by a native English speaker.
Reviewer #1:
Major comments
Author has presented an overview of the research on TAGE mostly from the formation perspective and its impact on onset/progression of lifestyle-related diseases (LSRD). While the review is well written, there are some important aspects that need to be considered by the author for a more comprehensive review.
- Authors needs to describe mechanisms by which TAGE drives LSRD. Does TAGE require engagement with its receptor to mediate cellular and molecular events to drive LSRD? Does receptor fo advanced glycation endproduct mediate some of TAGE actions in LSRD?
(A) Thank you for your valuable comments. Regarding the questions you raised, I have explained these details in Biomolecules (2021) and Cells (2022). In addition, I have provided an additional explanation of the TAGE-RAGE axis on page 5, lines 164–189.
Many cell types that possess pathways for the generation of GA from glucose and/or fructose may produce intracellular TAGE and subsequently induce cell death. TAGE leaks from cells and interact with RAGE to exert their effects on the surrounding cells. The TAGE-RAGE axis alters intracellular signaling, which up-regulates the expression of RAGE and contributes to LSRD.
- Figure 1 shows that polyol pathway (aldose reductase pathway) is, in part, responsible for generation of TAGE. Authors should discuss preclinical and clinical studies that may have shown that blocking this pathway reduces TAGE and consequent LSRD.
(A) The reduction in blood TAGE levels by atorvastatin administration, which inhibits the expression of fructokinase, a key enzyme in fructose metabolism, is described on page 12, lines 454–458. As you commented, I think further clinical research is necessary in the future.
The administration of atorvastatin to Sprague–Dawley male rats that had consumed a liquid fructose solution (10% w/v) abrogated the inflammatory and metabolic changes induced in the liver by fructose. These beneficial effects were attributed to the anti-inflammatory activity of atorvastatin and its downregulation of the hepatic expression of fructokinase, which inhibits fructose metabolism in the liver [Toxicol. Appl. Pharmacol. 2011; 251: 32-40 (PMID: 21122807)].
Reduced synthesis of GA leads to a drop in TAGE synthesis. Atorvastatin can reduce blood TAGE levels in a cholesterol-lowering-independent manner without altering glucose metabolism.
- Are TAGE degradation pathways altered in LSRD? Perhaps, a brief description on the balance between TAGE production vs degradation in the context of LSRD would be helpful.
(A) Further details regarding the formation and degradation of TAGE are the subject of future research.
Detail comments
Nothing specific to report.
Reviewer 2 Report
Toxic advanced glycation end-products (TAGE) are probably related to onset/progression of lyfestile-related disease.
The Author published other articles on this issue, very similar.
Please verify if this is a repeated manuscript ( same figures, same statements...)
The topic is very interesting but the originality is doubtful given the other very similar publications
Author Response
I would like to thank the Reviewers for their careful consideration of my manuscript. The comments were very helpful and I hope that my point-by-point responses are satisfactory, as shown in red. Moreover, the revised manuscript has been proofread by a native English speaker.
Reviewer #2:
Major comments
Toxic advanced glycation end-products (TAGE) are probably related to onset/progression of lifestyle-related disease. The Author published other articles on this issue, very similar.
Please verify if this is a repeated manuscript (same figures, same statements...)
(A)Considering your observation, I removed Figures similar to those in previous papers.
The overall similarity of the papers is about 30% (within 7% individually). The overlapping areas contain a lot of technical terminology, which makes the degree of overlap appear high.
Detail comments
The topic is very interesting but the originality is doubtful given the other very similar publications.
(A) Thank you for your valuable comments. TAGE is an AGE that our group discovered, and I believe that TAGE theory will have a significant impact. This paper provides an updated and comprehensive overview of the associations among TAGE and LSRD, which represents a unique contribution to the literature.
Reviewer 3 Report
In this manuscript, “Toxic AGEs (TAGE) cause lifestyle-related diseases” by Masayoshi Takeuchi reviews the TAGE theory provides novel insights into LSRD and is expected to elucidate new targets for many diseases. Although, the author had collected a lot literature, this work still has to revise.
1. The abstract should be revised.
2. The first paragraph of the introduction seems duplicated from the abstract.
3. Citations and copyrights of some Figures should be added or granted.
Author Response
I would like to thank the Reviewers for their careful consideration of my manuscript. The comments were very helpful and I hope that my point-by-point responses are satisfactory, as shown in red. Moreover, the revised manuscript has been proofread by a native English speaker.
Reviewer #3:
Major comments
In this manuscript, “Toxic AGEs (TAGE) cause lifestyle-related diseases” by Masayoshi Takeuchi reviews the TAGE theory provides novel insights into LSRD and is expected to elucidate new targets for many diseases. Although, the author had collected a lot literature, this work still has to revise.
(A) Thank you for your valuable comments. Following your suggestion, I revised the paper.
Detail comments
- The abstract should be revised.
(A) Following your suggestion, I have revised the first half of the Abstract (highlighted in yellow, page 1, lines 7–12).
- The first paragraph of the introduction seems duplicated from the abstract.
(A) I have revised the Abstract (highlighted in yellow, page 1, lines 7–12).
- Citations and copyrights of some Figures should be added or granted.
(A) Accordingly, I revised Figure 1 and deleted Figure 5.
Reviewer 4 Report
1. Please describe the fructolysis, glycolysis, and polyol pathways of AGEs formation in detail in 3. Overview of AGEs Formation Pathway in the Body.
2. TAGE is involved in DM, NASH, CVD, AD, cancer, and infertility. However, authors mostly explained the production of AGEs by hyperglycemia in DM. Please add how AGEs are produced in other diseases.
3. Please add the subtypes of RAGE and their role in the toxic effects of AGEs.
4. Please add the therapeutic drugs of TAGE in LSRD.
1. Figure 5 is too simple, possible subtypes of TAGE should be added.
Author Response
I would like to thank the Reviewers for their careful consideration of my manuscript. The comments were very helpful and I hope that my point-by-point responses are satisfactory, as shown in red. Moreover, the revised manuscript has been proofread by a native English speaker.
Reviewer #4:
Major comments
- Please describe the fructolysis, glycolysis, and polyol pathways of AGEs formation in detail in 3. Overview of AGEs Formation Pathway in the Body.
(A) Thank you for your valuable comments. I added an explanation (highlighted in yellow, page 3, lines 105–116).
- TAGE is involved in DM, NASH, CVD, AD, cancer, and infertility. However, authors mostly explained the production of AGEs by hyperglycemia in DM. Please add how AGEs are produced in other diseases.
(A) Following your suggestion, I added a short description (yellow highlighted page 3, lines 117–122).
- Please add the subtypes of RAGE and their role in the toxic effects of AGEs.
(A) Accordingly, I added an explanation (highlighted in yellow, page 5, lines 164–189).
- Please add the therapeutic drugs of TAGE in LSRD.
(A) I added an explanation (yellow highlighted page 12, lines 441–474).
Detail comments
- Figure 5 is too simple, possible subtypes of TAGE should be added.
(A) I deleted Figure 5.
Round 2
Reviewer 2 Report
The modified version of the manuscript is more complete and it is suitable for publication.
Abstract and introduction has been correctly modified and the new statements are very interesting
Author Response
Major comments
The modified version of the manuscript is more complete and it is suitable for publication.
Detail comments
Abstract and introduction has been correctly modified and the new statements are very interesting.
(A) Thank you for your valuable comments.
Reviewer 3 Report
Citations and copyrights of some Figures should be added or granted.
Citations and copyrights of some Figures should be added or granted.
Author Response
Major comments
Citations and copyrights of some Figures should be added or granted.
Detail comments
Citations and copyrights of some Figures should be added or granted.
(A) Thank you for your valuable comments. The figures shown here are original figures that I created.